# Enhancing Motion in Text-to-Video Generation with Decomposed Encoding and Conditioning

**Penghui Ruan**[1,2], **Pichao Wang**[3]*, **Divya Saxena**[1], **Jiannong Cao**[1†], **Yuhui Shi**[2†]

[1]The Hong Kong Polytechnic University, Hong Kong, China
[2]Southern University of Science and Technology, Shenzhen, China
[3]Amazon, Seattle, United States
penghui.ruan@connect.polyu.hk, pichaowang@gmail.com
{divsaxen, csjcao}@comp.polyu.edu.hk, shiyh@sustech.edu.cn

## Abstract

Despite advancements in Text-to-Video (T2V) generation, producing videos with realistic motion remains challenging. Current models often yield static or minimally dynamic outputs, failing to capture complex motions described by text. This issue stems from the internal biases in text encoding, which overlooks motions, and inadequate conditioning mechanisms in T2V generation models. To address this, we propose a novel framework called DEcomposed MOtion (DEMO), which enhances motion synthesis in T2V generation by decomposing both text encoding and conditioning into content and motion components. Our method includes a content encoder for static elements and a motion encoder for temporal dynamics, alongside separate content and motion conditioning mechanisms. Crucially, we introduce text-motion and video-motion supervision to improve the model's understanding and generation of motion. Evaluations on benchmarks such as MSR-VTT, UCF-101, WebVid-10M, EvalCrafter, and VBench demonstrate DEMO's superior ability to produce videos with enhanced motion dynamics while maintaining high visual quality. Our approach significantly advances T2V generation by integrating comprehensive motion understanding directly from textual descriptions. Project page: https://PR-Ryan.github.io/DEMO-project/

## 1 Introduction

The field of Text-to-Video (T2V) generation [21, 46, 7, 8, 19, 56, 26, 74, 3, 59, 70] has seen significant advancements, especially with the advent of diffusion models. These models have demonstrated impressive capabilities in generating visually appealing videos from textual descriptions. However, a persistent challenge remains: generating videos with realistic and complex motions. Most existing T2V models produce outputs that resemble static animations or exhibit minimal camera movement, falling short of capturing the intricate motions described in textual inputs [21, 46, 7, 19, 56, 74].

This limitation arises from two primary challenges. The first challenge is the inadequate motion representation in text encoding. Current T2V models utilize large-scale visual-language models (VLMs), such as CLIP [40], as text encoders. These VLMs are highly effective at capturing static elements and spatial relationships but struggle with encoding dynamic motions. This is primarily due to their training focus, which biases them towards recognizing nouns and objects [35], while verbs and actions are less accurately represented [17, 69, 38]. The second challenge is the reliance on spatial-only text conditioning. Existing models often extend Text-to-Image (T2I) generation

---

*The work does not relate to author's position at Amazon.
†Corresponding authors.

38th Conference on Neural Information Processing Systems (NeurIPS 2024).

techniques to T2V tasks [21, 46, 7, 19, 56, 74, 3], applying text information through spatial cross-attention on a frame-by-frame basis. While effective for generating high-quality static images, this approach is insufficient for videos, where motion is a critical component that spans both spatial and temporal dimensions. A holistic approach that integrates text information across these dimensions is essential for generating videos with realistic motion dynamics.

Recent efforts to address these challenges have involved incorporating additional control signals such as sketches [15], strokes [9, 23, 60, 51, 68], database samples [72], depth maps [33], and human poses [71, 6, 12], reference videos [62, 73, 34, 65], and bounding boxes [55] into the T2V generation process. These signals are derived either from reference videos or pre-trained motion generation models [36, 30]. While these approaches improve motion synthesis, they depend on external references or pre-trained models, which may not always be practical. Moreover, they introduce complexity and potential inefficiencies, as they require separate handling of additional data sources.

To address these challenges, we introduce Decomposed Motion (DEMO), a novel framework designed to enhance motion synthesis in T2V generation. DEMO adopts a comprehensive approach by decomposing both text encoding and conditioning processes into content and motion components. Addressing the first challenge, DEMO decomposes text encoding into content encoding and motion encoding processes. The content encoding focuses on object appearance and spatial layout, capturing static elements such as "a girl" and "the road" in the scenario "A girl is walking to the left on the road." Meanwhile, the motion encoding captures the essence of object movement and temporal dynamics, interpreting actions like "walking" and directional cues like "to the left." This separation allows the model to better understand and represent the dynamic aspects of the described scenes. Regarding the second challenge, DEMO decomposes the text conditioning process into content and motion dimensions. The content conditioning module integrates spatial embeddings into the video generation process on a frame-by-frame basis, ensuring that static elements are accurately depicted in each frame. In contrast, the motion conditioning module operates across the temporal dimension, infusing dynamic motion embeddings into the video. This separation enables the model to capture and reproduce complex motion patterns described in the text. Moreover, DEMO incorporates novel text-motion and video-motion supervision techniques to enhance the model's understanding and generation of motion. Text-motion supervision aligns cross-attention maps with the temporal changes observed in ground truth videos, guiding the model to focus on motion information. Video-motion supervision constrains the predicted video latent to mimic the motion patterns of real videos, promoting the generation of coherent and realistic motion dynamics. These supervision techniques ensure that the model not only generates visually appealing videos but also renders the intricate motions described in the text.

To validate our framework, we conduct extensive experiments on several benchmarks, including MSR-VTT [66], UCF-101 [50], WebVid-10M [1], EvalCrafter [31], and VBench [24]. DEMO achieves substantial improvements in metrics related to motion dynamics and visual fidelity, indicating its superior capability to generate videos that are both visually appealing and dynamically accurate.

## 2 Related Work

**T2V Generation.** The T2V domain has made substantial strides, building on the progress in T2I generation. The first T2V model, VDM [21], introduces a space-time factorized U-Net for temporal modeling, training on both images and videos. For high-definition videos, models like ImagenVideo [19], Make-A-Video [46], LaVie [59], and Show-1 [70] use cascades of diffusion models with spatial and temporal super-resolution. MagicVideo [74], Video LDM [4], and LVDM [16] apply latent diffusion for video, working in a compressed latent space. VideoFusion [32] separates video noise into base and residual components. ModelScopeT2V uses 1D convolutions and attention to approximate 3D operations. Stable Video Diffusion (SVD) [3] divides the process into T2I pre-training, video pre-training, and fine-tuning and demonstrate the necessity of a well-curated high-quality pretraining dataset for developing a strong base model. Despite these advancements, the generated videos still exhibit limited motion dynamics, often appearing largely static with minimal motion, highlighting an ongoing challenge in achieving dynamic and realistic motions.

**T2V Generation with Rich Motion.** Generating video with rich motion is still an open challenge in the field of T2V generation. Existing works [15, 9, 23, 60, 51, 68, 72, 33, 71, 6, 12, 62, 73, 34,

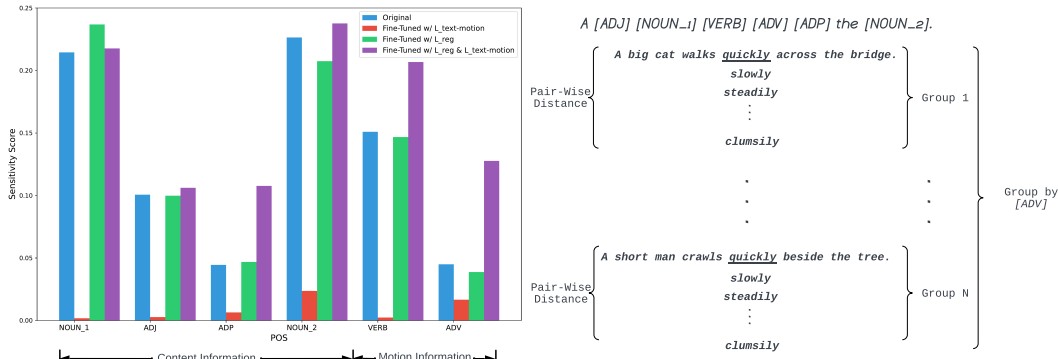

Figure 1: **Our Pilot Study**. We generated a set of prompts (262144 in total) following a fixed template, grouping them according to the different parts of speech (POS). These grouped texts are then passed into the CLIP text encoder, and we calculate the sensitivity as the average sentence distance within each group. As shown on the left-hand side, compared to POS representing content, CLIP is less sensitive to POS representing motion. (Results are consistent across different templates and different sets of words within each POS. Further details can be found in the appendix.)

65, 55] address this challenge by incorporating additional control signals that inherently contain rich motion information. Tune-A-Video [65] proposes spatial-temporal self-attention into the T2I backbone and trains the model on a single reference video. The model thus learns to generate new videos with motions specified by the reference video. Materzynska *et al.* [34] follow the idea of T2I customization [13, 42, 27] to fine-tune the model and a specific text token on a small set of reference videos. The model can then recontextualize with that learned token to generate new videos with specific motions. DreamVideo [62] further customizes both the appearances and motions given reference images and videos. MotionDirector [73] proposes a dual-path Low-Rank Adaptations [22] to decouple the motions and appearances residing in the reference videos. MotionCtrl [60] incorporates object trajectories and camera poses into the T2V generation by conditioning them in the convolution and temporal transformer layers, respectively. Contrasting with these approaches, DEMO prioritizes the generation of videos that exhibit significant motions derived solely from textual descriptions without relying on additional signals.

## 3 Method

**Latent Video Diffusion Models (LVDMs).** LVDMs build on the diffusion models [48, 20] by training a 3D U-Net as the noise predictor, where a VQ-VAE [37] or a VQ-GAN [11] is employed to compress the video into low-dimensional latent space. The 3D U-net consists of down-sample, middle, and up-sample blocks. Each of these blocks comprises multiple convolution layers augmented by spatial and temporal transformers. The spatial transformer consists of spatial self-attention, spatial cross-attention, and feed-forward layers. The temporal transformer consists of temporal self-attention and feed-forward layers. The 3D U-Net is trained with a text encoder to minimize the noise-prediction loss in the latent space given as follows:

$$\mathcal{L}_{\text{diffusion}} = \mathbb{E}_{t,z_0,\epsilon \sim \mathcal{N}(0,1),p}[||\epsilon - \epsilon_\theta(z_t, t, \mathcal{E}(p))||_2^2] \tag{1}$$

where $z$ is the video latent corresponding to $x$ in the pixel space, $t$ is the time step, $\mathcal{E}$ is a text encoder, $p$ is a text prompt, and $\epsilon$ is noise sampled from Gaussian distribution. $z_t$ is noisy $z_0$ after $t$ steps diffusion forward process given by:

$$z_t = \sqrt{\bar{\alpha}_t}z_0 + \sqrt{1 - \bar{\alpha}_t}\epsilon, \ \bar{\alpha}_t = \prod_{s=1}^{t} \alpha_s \tag{2}$$

where $\alpha_t$ is a pre-defined noise schedule.

### 3.1 Decomposed Text Encoding

As shown in our pilot study in Figure 1, the CLIP text encoder can distinguish different motions, but it is not as sensitive to motion as it is to content. Consequently, the text encoding focuses more

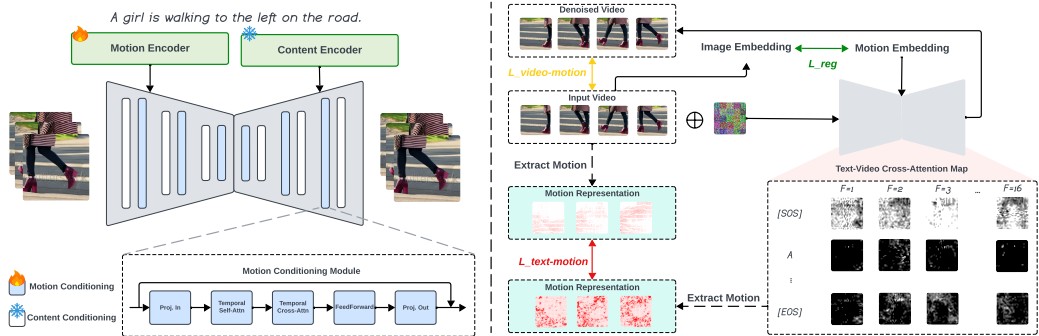

Figure 2: **Overview of DEMO Training**. As shown in the left-hand side, DEMO incorporate dual text encoding and text conditioning (for simplicity, other layers in the UNet are omitted). As shown in the right-hand side, during training, the $\mathcal{L}_{\text{text-motion}}$ is used to enhance motion encoding, the $\mathcal{L}_{\text{reg}}$ is used to avoid catastrophic forgetting, the $\mathcal{L}_{\text{video-motion}}$ is to enhance motion integration. The **snowflakes** and **flames** denote frozen and trainable parameters, respectively.

on content encoding rather than motion encoding. To preserve the generalization ability of the T2V generation model, we retain the original text encoder, referring to it as the content encoder (denoted as $\mathcal{E}_c$). Additionally, we introduce a new text encoder, referred to as the motion encoder (denoted as $\mathcal{E}_m$), which is specifically designed to capture object movement and temporal dynamics in textual descriptions (as shown in the left-hand side of Figure 2). We initialize our motion encoder from a CLIP text encoder and then fine-tune it using specialized text-motion supervision, as described below.

**Text-Motion Supervision.** Research [27, 28, 61] has shown that cross-attention maps represent the structure of visual content. The cross-attention operation can be viewed as a projection of text information into the visual structure domain. With this understanding, we aim to shift the text encoder's focus more toward motion information by constraining the temporal changes of cross-attention maps to closely mimic those observed in ground truth videos, as illustrated by the **red** line in Figure 2. Formally, given a noisy video latent $z_t$ at time step $t$ and a text prompt $p$, the cross-attention maps $\mathcal{A}^i \in \mathbb{R}^{H^i \times W^i \times F \times S}$, where $H^i$ and $W^i$ are the height and width of video latent at the $i$th cross-attention layer, $F$ is the number of frames, $S$ is the sequence length, for a cross-attention layer $i$ are defined as follows:

$$\mathcal{A}^i = \frac{1}{N} \sum_n^N \text{softmax}\left(\frac{Q^{(n)}(K^{(n)})^T}{\sqrt{d_n}}\right) \tag{3}$$

$$Q^{(n)} = W_Q^{(n)} \cdot z_t, K^{(n)} = W_K^{(n)} \cdot \mathcal{E}_m(p) \tag{4}$$

where $W_Q$ and $W_K$ are projection matrices for query and key, $i \in \{1, 2, ...M\}$ is layer index, $n \in \{1, 2, ..., N\}$ represents each head in multi-head cross-attention, and $d_n$ is the dimension of each head.

We empirically find that the cross-attention maps corresponding to the "[eos]" token, which aggregate the whole sentence's semantics, play a pivotal role in generating motion. This aligns with the understanding that motion is a global concept and cannot be captured by a single word. For instance, phrases like "A baby/dog is walking/running forwards/backward." demonstrate that different combinations of words can result in significantly different motions. Hence, we focus on the cross-attention maps related to the "[eos]" token and constrain them to mimic the motion patterns observed in the ground truth videos. This approach forms the basis of our text-motion loss, defined as follows:

$$\mathcal{L}_{\text{text-motion}} = -\mathbb{E}_{t,x_0,\epsilon \sim \mathcal{N}(0,1),p} \left[ \frac{1}{M} \sum_{i=1}^{M} \cos(\phi(\mathcal{A}^i_{[eos]}), \phi(x_0)) \right] \tag{5}$$

where $\phi$ is a function to extract motion dynamics from a video. In our case, we use optical flow to represent the motion dynamics (noting that optical flow is only used during training; during

inference, we use only the text prompt as input). In light of the potential scale differences between the cross-attention maps and video pixel values, we compute the cosine similarity between them. Additionally, for cross-attention at different spatial resolutions, we downsample the ground truth video to match the spatial resolution of the cross-attention maps.

**Regularization.** Recall that CLIP [40] is trained with a contrastive learning objective to match texts and images from a group of text-image pairs. However, directly fine-tuning the CLIP text encoder with Equation 1 and Equation 5, which differ significantly from the original contrastive learning objective, can easily lead to catastrophic forgetting [29]. To mitigate this, we introduce a regularization term in the fine-tuning objective to preserve its generalization ability. Specifically, we penalize the text embedding if it diverges from the corresponding image embedding, maintaining alignment with the original CLIP contrastive learning objective, as illustrated by the **green** line in Figure 2. The regularization loss is defined as follows:

$$\mathcal{L}_{\text{reg}} = -\mathbb{E}_{x_0,p}\left[cos(\mathcal{E}_m(p), \mathcal{E}^{img}(x_0^{F/2}))\right] \tag{6}$$

where $\mathcal{E}^{img}$ represents the CLIP image encoder. Given that there is only one text prompt for the entire video, we select medium frame $x_0^{F/2}$ and compute its image embedding.

## 3.2 Decomposed Text Conditioning

DEMO employs separate content conditioning and motion conditioning modules to incorporate content and motion information. To preserve the generative capabilities of our base model, we maintain the original text conditioning module, referred to here as the content conditioning module. We then strategically introduce a novel temporal transformer, referred to as the motion conditioning module (detailed structure shown in Figure 2), to incorporate motion information along the temporal axis. To encourage the motion conditioning module to generate and render motion dynamics, we train this module under video-motion supervision, as described below.

**Video-Motion Supervision.** Recall that at each diffusion denosing step $t$, we can obtain the predicted $\hat{z}_{0,t}$ at time step $t$, which is given by:

$$\hat{z}_{0,t}(t, z_t, \mathcal{E}_m(p), \mathcal{E}_c(p)) = \frac{z_t - \sqrt{1 - \bar{\alpha}_t}\epsilon_\theta(z_t, t, \mathcal{E}_m(p), \mathcal{E}_c(p))}{\sqrt{\bar{\alpha}_t}} \tag{7}$$

This predicted $\hat{z}_{0,t}$ encapsulates the motion information in the video domain. We then prioritize the motion generation by constraining the predicted $\hat{z}_{0,t}$ to mimic the motion pattern in the real video, as illustrated by the yellow line in Figure 2. We define our video-motion loss as follows:

$$\mathcal{L}_{\text{video-motion}} = \mathbb{E}_{t,z_0,\epsilon\sim\mathcal{N}(0,1)}\|\Phi(z_0) - \Phi(\hat{z}_{0,t})\|_2^2 \tag{8}$$

where $\Phi$ is a function to extract motion features from a video. Given that $\mathcal{L}_{\text{diffusion}}$ is a pixel-wise denoising loss (whether raw pixel or latent pixel), choosing $\Phi$ as a general motion representation that is not in pixel space may lead to conflicting objectives due to the differing representation spaces. Instead, we choose $\Phi$ as the consecutive frame difference defined as follows:

$$\Phi(z_0) = z_0^{2:F} - z_0^{1:F-1} \tag{9}$$

where $z_0^{2:F}$ denotes the video latent from frame 2 to frame $F$, and $z_0^{1:F-1}$ denotes the video latent from frame 1 to frame $F - 1$.

## 3.3 Joint Training

Our final loss is a weighted combination of $\mathcal{L}_{\text{text-motion}}$, $\mathcal{L}_{\text{reg}}$, $\mathcal{L}_{\text{video-motion}}$, and original diffusion loss $\mathcal{L}_{\text{diffusion}}$ as follows:

$$\mathcal{L} = \mathcal{L}_{\text{diffusion}} + \alpha\mathcal{L}_{\text{text-motion}} + \beta\mathcal{L}_{\text{reg}} + \gamma\mathcal{L}_{\text{video-motion}} \tag{10}$$

where $\alpha$, $\beta$, and $\gamma$ are scaling factors to balance different loss terms.

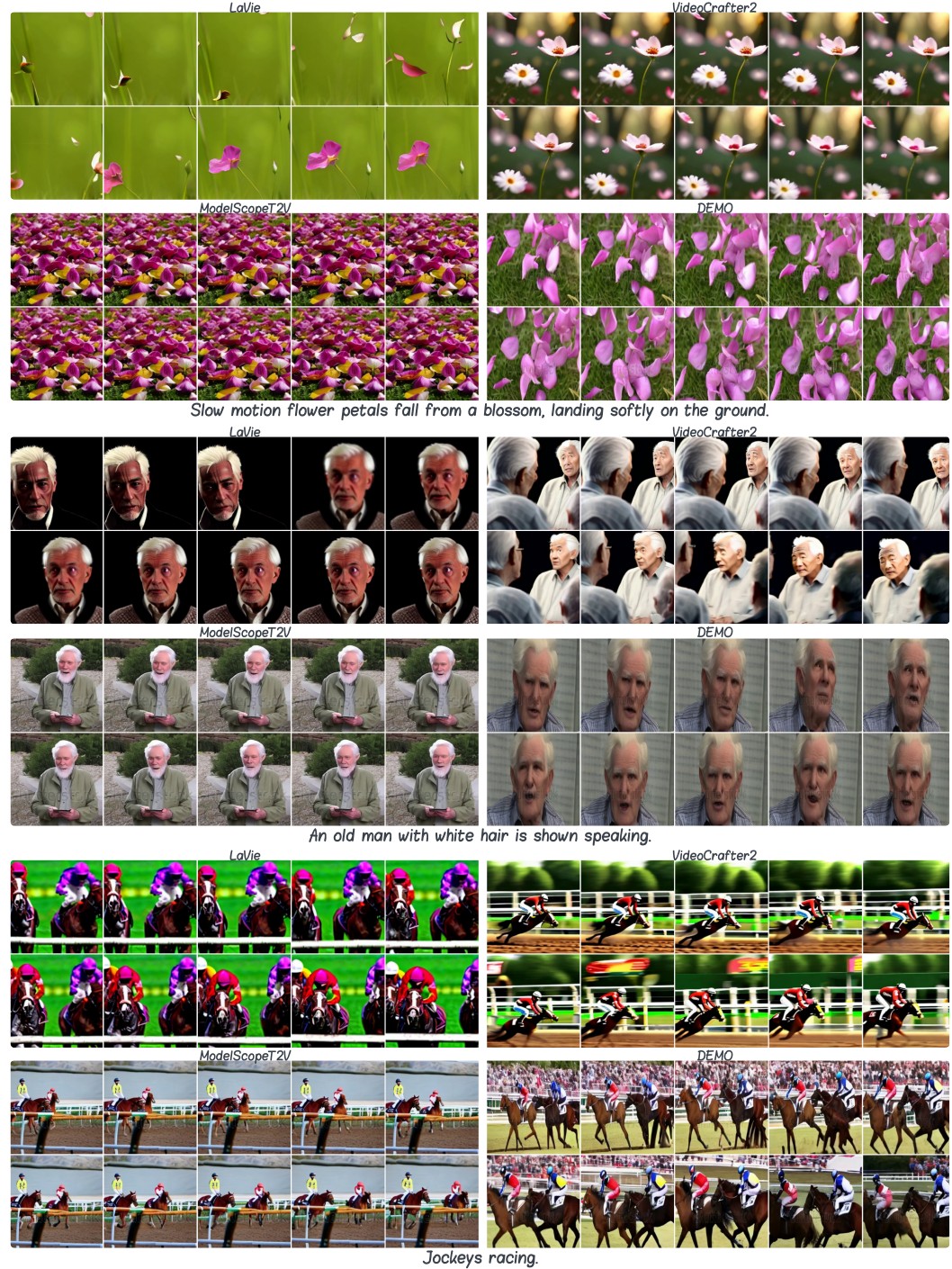

Figure 3: **Qualitative Comparison.** Each video is generated with 16 frames. We display frames 1, 2, 4, 6, 8, 10, 12, 14, 15, and 16, arranged in two rows from left to right. Full videos are available in the supplementary materials.

# 4 Experiments

## 4.1 Implementation Details

To rule out potential dataset bias, we use the same training dataset as our base model ModelScopeT2V. Specifically, we use WebVid-10M [1], a large-scale dataset of short videos with textual descriptions as our fine-tuning dataset. The training details and hyperparameters can be found in the Appendix.

## 4.2 Qualitative Evaluations

In this subsection, we conduct a qualitative comparison among LaVie [59], VideoCrafter2 [8], ModelScopeT2V [56], and DEMO. For a fair comparison, we use the same seed for each of these methods. The comparative analysis is illustrated in Figure 3, where we showcase examples generated by these methods. Upon examination, it is evident that these models are capable of producing high-quality videos. However, a notable distinction arises in the dynamic representation of motion within the generated videos. The ModelScopeT2V model, while visually appealing, predominantly generates static scenes. For instance, in the scenario described as "Slow motion flower petals fall from a blossom, landing softly on the ground" (the first example in Figure 3), the video generated by ModelScopeT2V captures the petals landing on the ground but lacks the motion of the petals falling. In contrast, DEMO significantly outperforms by capturing the essence of motion, producing a video where the petals fall slowly and gently to the ground. Similarly, LaVie demonstrates a similar issue, as illustrated in the third example, where the jockeys remain largely static. VideoCrafter2 exhibits relatively large motion dynamics but suffers from motion blur, as shown in the third example. Conversely, DEMO vividly captures the jockeys racing, thereby providing a more realistic representation. This underscores the advanced capability of DEMO to generate videos that not only visually represent a scene but also dynamically encapsulate the ongoing motion.

## 4.3 Quantitative Evaluations

Table 1: Results of zero-shot T2V generation on MSR-VTT (Evaluation protocol comparison can be found in the appendix).

| Model | FID ($\downarrow$) | FVD ($\downarrow$) | CLIPSIM ($\uparrow$) |
|---|---|---|---|
| MagicVideo [74] | - | 1290 | - |
| Make-A-Video [46] | 13.17 | - | 0.3049 |
| Show-1 [70] | 13.08 | 538 | 0.3072 |
| Video LDM [4] | - | - | 0.2929 |
| LaVie [59] | - | - | 0.2949 |
| PYoCo [14] | 10.21-9.73 | - | - |
| VideoFactory [58] | - | - | 0.3005 |
| EMU VIDEO [45] | - | - | - |
| SVD [3] | - | - | - |
| ModelScopeT2V[3] [56] | 14.89 | 557 | 0.2941 |
| ModelScopeT2V fine-tuned | 13.80 | 536 | 0.2932 |
| **DEMO** | **11.77** | **422** | **0.2965** |

Table 2: Results of zero-shot T2V generation on UCF-101 (Evaluation protocol comparison can be found in the appendix).

| Model | IS ($\uparrow$) | FVD ($\downarrow$) |
|---|---|---|
| MagicVideo [74] | - | 655.00 |
| Make-A-Video [46] | 33.00 | 367.23 |
| Show-1 [70] | 35.42 | 394.46 |
| Video LDM [4] | 33.45 | 550.61 |
| LaVie [59] | - | 526.30 |
| PYoCo [14] | 47.76 | 355.19 |
| VideoFactory [58] | - | 410.00 |
| EMU VIDEO [45] | 42.70 | 606.20 |
| SVD [3] | - | 242.02 |
| ModelScopeT2V [56] | **37.55** | 628.17 |
| ModelScopeT2V fine-tuned | 37.21 | 612.53 |
| **DEMO** | 36.35 | **547.31** |

**Zero-shot T2V Generation on MSR-VTT.** We evaluate the performance of our model on the MSR-VTT [66] test set by calculating FID [18], FVD [54, 39], and CLIPSIM [63] metrics. For FID and FVD, in alignment with prior studies [56], we randomly sample 2048 videos and one prompt for each video from the test set. For CLIPSIM, we follow previous works [46, 64, 56] and use nearly 60k sentences from the entire test set to generate videos. As illustrated in Table 1, DEMO demonstrates notable advancements over the ModelScopeT2V baseline in terms of video quality metrics. Specifically, DEMO achieves an FID score of 11.77, showing marked improvement in individual frame quality compared to the baseline score of 14.89. For FVD, DEMO achieves a score of 422 compared to the baseline of 557, indicating improved overall video quality. It is important to note that the FVD is calculated using an I3D model pre-trained on the Kinetics-400 dataset [5] for action recognition. By computing the FVD over its logits, this metric not only reflects visual quality but also emphasizes motion quality in video generation. Additionally, DEMO improves the CLIPSIM

---

[3]Results reproduced from our own evaluation.

Table 3: Results of T2V generation on WebVid-10M (Val).

| Model | FID ($\downarrow$) | FVD ($\downarrow$) | CLIPSIM ($\uparrow$) |
|---|---|---|---|
| ModelScopeT2V | 11.14 | 508 | 0.2986 |
| ModelScopeT2V fine-tuned | 10.53 | 461 | 0.2952 |
| DEMO | **9.86** | **351** | **0.3083** |

Table 4: Results of zero-shot T2V generation on EvalCrafter.

| Model | Video Quality | | | Motion Quality | | |
|---|---|---|---|---|---|---|
| | VQA$_A$ ($\uparrow$) | VQA$_T$ ($\uparrow$) | IS ($\uparrow$) | Action Score ($\uparrow$) | Motion AC-Score ($\uparrow$) | Flow Score ($\uparrow$) |
| ModelScopeT2V | 15.12 | **16.88** | 14.60 | 75.88 | 44 | 2.51 |
| ModelScopeT2V fine-tuned | 15.89 | 16.39 | 14.92 | 74.23 | 40 | 2.72 |
| DEMO w/o $\mathcal{L}_{\text{video-motion}}$ | 18.78 | 15.12 | 17.13 | 76.20 | 48 | 3.11 |
| DEMO | **19.28** | 15.65 | **17.57** | **78.22** | **58** | **4.89** |

score from 0.2941 to 0.2965, further demonstrating its superior ability to generate high-quality videos that are well-aligned with their textual descriptions.

**Zero-shot T2V Generation on UCF-101.** For UCF-101 [50], we report the IS [43] and FVD on the 101 action classes. For IS and FVD, we follow previous works [4, 14] to generate 100 videos for each of the 101 classes. We directly use the class names as prompts. As shown in Table 2, compared with baseline ModelScoprT2V, we improve the FVD from 628.17 to 547.31. However, we observed a slight decrease in IS, which may be attributed to the limited textual information provided by UCF-101 class names, such as "baby crawling" and "cliff diving." These prompts primarily suggest motion, and our model, optimized to emphasize this motion, may have over-focused on this limited information. This overemphasis potentially limited the diversity of generated content, lowering the IS.

**T2V Generation on WebVid-10M (Val).** For WebVid-10M [1], we perform T2V generation on the validation set. As shown in Table 3, we evaluate the FID, FVD, and CLIPSIM, where we randomly sample 5K text-video pairs from the validation set. Our model achieves an FID score of 9.86, an FVD score of 351, and a CLIPSIM score of 0.3083. These outcomes underscore our framework's substantial enhancement of video quality.

**Zero-shot T2V Generation on EvalCrafter.** EvalCrafter [31] provides 700 diverse prompts across categories like human, animal, objects, and landscape, each with a scene, style, and camera movement description. For our evaluation, we generate one video for each of the 700 text prompts. As shown in Table 4, we have obtained significant improvement over the baseline ModelScopeT2V in both video quality and motion quality. In terms of video quality, DEMO enhances both the Video Quality Assessment for Aesthetics (VQA$_A$) and the IS, albeit with a slight decrease in the Video Quality Assessment for Technical Quality (VQA$_T$). For motion quality, EvalCrafter uses three metrics: Action-Score, Flow-Score, and Motion AC-Score. The Action-Score, based on the VideoMAE V2 model [57] and MMAction2 toolbox, measures action recognition accuracy on Kinetics-400 classes, with higher scores indicating better human action recognition. Flow-Score and Motion AC-Score, derived from RAFT model [53] optical flows, evaluate motion dynamics. The Flow-Score measures the general motion dynamics by calculating the average magnitude of optical flow in the video, while the Motion AC-Score assesses how well the motion dynamics align with the text prompt. For motion quality, our model surpasses the baseline across all metrics (Action-Score, Flow-Score, and Motion AC-Score), showcasing DEMO's superior ability to generate videos characterized by better motion quality and higher motion dynamics.

**Zero-shot T2V Generation on VBench.** VBench [24] is a comprehensive benchmark to evaluate video quality. In our evaluation of VBench, we focus specifically on motion quality. We report on

Table 5: Results of zero-shot T2V generation on VBench.

| Model | Motion Dynamics ($\uparrow$) | Human Action ($\uparrow$) | Temporal Flickering ($\uparrow$) | Motion Smoothness($\uparrow$) |
|---|---|---|---|---|
| ModelScopeT2V | 62.50 | 90.40 | 96.02 | 96.19 |
| ModelScopeT2V fine-tuned | 63.75 | 90.40 | **96.35** | **96.38** |
| DEMO | **68.90** | **90.60** | 94.63 | 96.09 |

four key metrics: Motion Dynamics, Human Action, Temporal Flickering, and Motion Smoothness. As shown in Table 5, DEMO significantly improves motion dynamics from 62.50 to 68.90. However, we observed only a slight improvement in human action recognition. This indicates that while our model enhances the richness and complexity of motion, it provides limited benefit in improving the accuracy of human action representation. Additionally, we note slight decreases in temporal flickering and motion smoothness. This observation aligns with findings from the VBench paper, which suggest that increased motion dynamics can conflict with temporal flickering and motion smoothness.

## 4.4 Ablation Studies

**Impact of $\mathcal{L}_{\text{reg}}$ and $\mathcal{L}_{\text{text-motion}}$.** As shown in Figure 1, we compute the sensitivity of our motion encoder with different loss combinations. The red columns indicate the motion encoder with $\mathcal{L}_{\text{text-motion}}$ only completely loses its ability to distinguish different tokens, either motion or content, indicating a serious catastrophic forgetting where the model loses its original knowledge. The green columns show that fine-tuning the motion encoder with $\mathcal{L}_{\text{reg}}$ only preserves the model's generalization ability but does not increase the motion sensitivity. In contrast, the purple columns demonstrate that when training the motion encoder with both $\mathcal{L}_{\text{reg}}$ and $\mathcal{L}_{\text{text-motion}}$, the model gain increased sensitivity to tokens representing motion without losing sensitivity to tokens representing content.

**Impact of $\mathcal{L}_{\text{video-motion}}$.** To validate the effectiveness of our video-motion loss, we perform an ablation study on the EvalCrafter dataset. As shown in Table 4, without $\mathcal{L}_{\text{video-motion}}$, the model shows a slight improvement in motion quality compared to the baseline. This is because the motion encoder provides the model with enriched motion information for generation. However, without explicitly constraining the model to mimic realistic motion, it may still focus on generating high-quality individual frames rather than coherent video sequences with rich motion dynamics. By introducing video-motion loss, the model achieves significantly higher motion quality, demonstrating the importance of this loss in guiding the model in producing videos with enhanced motion dynamics.

Table 6: Ablation study on additional parameters in motion encoder.

| Benchmark | Metric | ModelScopeT2V | ModelScopeT2V fine-tuned | ModelScopeT2V + motion encoder | DEMO |
|---|---|---|---|---|---|
| MSRVTT | FID ($\downarrow$) | 14.89 | 13.80 | 13.98 | **11.77** |
| | FVD ($\downarrow$) | 557 | 536 | 552 | **422** |
| | CLIPSIM ($\uparrow$) | 0.2941 | 0.2932 | 0.2935 | **0.2965** |
| UCF-101 | IS ($\uparrow$) | 37.55 | 37.21 | **37.66** | 36.35 |
| | FVD ($\downarrow$) | 628.17 | 612.53 | 601.25 | **547.31** |
| WebVid-10M | FID ($\downarrow$) | 11.14 | 10.53 | 10.45 | **9.86** |
| | FVD ($\downarrow$) | 508 | 461 | 458 | **351** |
| | CLIPSIM ($\uparrow$) | 0.2986 | 0.2952 | 0.2967 | **0.3083** |
| EvalCrafter | VQA_A ($\uparrow$) | 15.12 | 15.89 | 16.21 | **19.28** |
| | VQA_T ($\uparrow$) | **16.88** | 16.39 | 16.34 | 15.65 |
| | IS ($\uparrow$) | 14.60 | 14.92 | 15.02 | **17.57** |
| | Action Score ($\uparrow$) | 75.88 | 74.23 | 75.20 | **78.22** |
| | Motion AC-Score ($\uparrow$) | 44 | 40 | 46 | **58** |
| | Flow Score ($\uparrow$) | 2.51 | 2.72 | 2.44 | **4.89** |
| Vbench | Motion Dynamics ($\uparrow$) | 62.50 | 63.75 | 63.50 | **68.90** |
| | Human Action ($\uparrow$) | 90.40 | 90.40 | 90.20 | **90.60** |
| | Temporal Flickering ($\uparrow$) | 96.02 | **96.35** | 95.45 | 94.63 |
| | Motion Smoothness ($\uparrow$) | 96.19 | **96.38** | 96.22 | 96.09 |

**Impact of additional parameters in motion encoder.** To rule out the effect of additional parameters introduce by motion encoder, we evaluated the effect of training with a CLIP text encoder on the overall model performance. We then compared three different variations: (1) the original ModelScopeT2V, (2) a fine-tuned version of ModelScopeT2V without additional motion encoder parameters, and (3) ModelScopeT2V with the motion encoder while maintaining its original training loss. As shown in Table 6, we observed that the performance of the model with the additional motion encoder parameters is comparable to the fine-tuned version without these extra parameters. This suggests that, without specific supervision or additional constraints, the effect of the added text encoder parameters is marginal. However, the DEMO model consistently outperforms all variations, demonstrating the effectiveness of our method in improving both video quality and text-video alignment.

**Efficiency Analysis.** To validate the efficiency of our proposed methods, we trained the baseline model for the same number of iterations and compared its performance with ours. As shown in Tables 1, 2, 3, 4, and 5, continuing to fine-tune the model results in only marginal improvements in

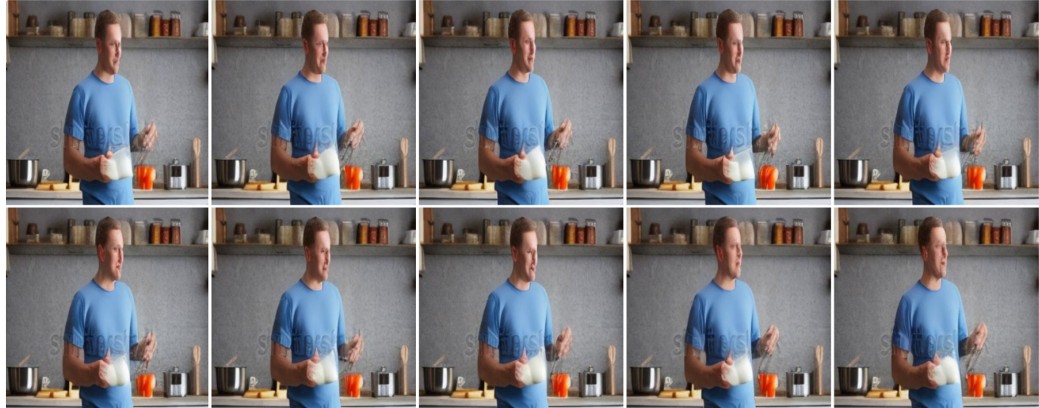

A man is standing in a kitchen talking and then a mixer and carton of milk are shown.

Figure 4: **Limitations**. DEMO does not support creating videos containing sequential motions specified by text. As shown in the example, two motions,"a man standing in a kitchen and talking" and "a mixer and a carton of milk are shown", appear simultaneously.

video quality. Additionally, we observed a slight degradation in CLIPSIM, indicating that further training may not benefit text-video alignment.

## 5   Limitations and Future Work

Despite DEMO's efficiency in enhancing motion synthesis without relying on additional signals, it faces significant challenges in generating different motions sequentially, as illustrated in Figure 4. These challenges likely stem from the text encoder's difficulty in comprehending the order of actions and the motion generation model's limited capability to generate different motions. A potential solution to this issue involves annotating each frame with a specific prompt and training the model on video clips of varying lengths rather than a fixed duration. We consider exploring this direction in our future work.

## 6   Broader Impacts

Our model achieves higher visual fidelity and motion quality, which can benefit various fields such as content creation and visual simulation. However, our model is fine-tuned on web data, specifically WebVid-10M [1]. As a result, the model may not only learn how to generate videos but also inadvertently learn societal biases present in the web data, which may include inappropriate or NSFW content. Potential post-processing steps, such as applying a video classifier to filter out undesirable content, could help mitigate this issue.

## 7   Conclusion

In this paper, we have presented DEMO, an innovative framework crafted to advance motion synthesis in T2V generation. By separating text encoding and text conditioning into distinct content and motion dimensions, DEMO facilitates the creation of static scenes and their dynamic evolution. To encourage our model to focus on motion encoding and motion generation, we propose novel text-motion and video-motion supervision. Our extensive evaluations across various benchmarks have illustrated DEMO's capability to significantly improve motion synthesis, showcasing its potential within the field. In future work, we plan to augment T2V datasets with more detailed descriptions and delve into advanced motion embedding techniques. By focusing on these areas, we aim to advance the frontiers of research in this dynamic and rapidly evolving domain.

## 8 Acknowledgement

This work is partially supported by National Science Foundation of China (No. 62250710682), Shenzhen Fundamental Research Program (No. JCYJ20200109141235597), NSFC/RGC Collaborative Research Scheme (No. CRS_PolyU501/23), HK RGC Theme-based Research Scheme (No. PolyU T43-513/23-N) and Research Grants Council of the Hong Kong Special Administrative Region, China (No. PolyU15205924). We also acknowledge the support from Research Institute for Artificial Intelligence of Things, The Hong Kong Polytechnic University, and Center for Computational Science and Engineering at Southern University of Science and Technology.

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

# 9 Appendix

## 9.1 Training Details and Hyperparameters

As shown in Table 7, we train DEMO using the Adam optimizer [25] with a OneCycle scheduler [47]. Specifically, the learning rate varies within the range of [0.00001, 0.00005], while the momentum oscillates between 0.85 and 0.99. We train our model using Deepspeed framework with stage 2 zero optimization and cpu offloading. DEMO is trained on 4 NVIDIA Tesla A100 GPUs with a batch size of 24 per GPU. DEMO takes $256\times256$ images as inputs and utilizes a VQGAN with a compression rate of 8 to encode images into a latent space of $32\times32$. DEMO is trained with 1000 diffusion steps. We set the classifier-free guidance scale as 9 with the probability of 0.1 randomly dropping the text during training. For inference, we use the DDIM sampler [49] with 50 inference steps.

Table 7: Training Hyperparameters

| Hyperparameter | | DEMO |
|---|---|---|
| | LDM | ✓ |
| | Compression Rate | 8 |
| | Latent Shape | $32\times32\times16$ |
| | Channels | 320 |
| U-net | Channel Multiplier | 1,2,4,4 |
| | Attention Resolutions | 16, 8, 4 |
| | Head Channels | 32 |
| | # of Parameters | 1.68B |
| | Dropout Rate | 0.1 |
| | Architecture | CLIP ViT-H/14 |
| Motion Encoder | Token Length | 77 |
| | Token Dimension | 1024 |
| | # of Parameters | 354.03M |
| | Proj In & Proj Out | Linear |
| | Normalization | GroupNorm 32 |
| | Activation | GEGLU |
| Motion Conditioning | Channels | 320 |
| | Attention Resolutions | 16, 8, 4 |
| | Head Channels | 32 |
| | # of Parameters | 238.32M |
| | DDPM Time Steps | 0, 1000 |
| | Optimizer | Adam |
| | Learning Rate | 0.00001, 0.00005 |
| | Scheduler | OneCycle Scheduler |
| Training | Classifier-free Guidance Scale | 9 |
| | Loss Weight $\alpha$ | 0.1 |
| | Loss Weight $\beta$ | 0.3 |
| | Loss Weight $\gamma$ | 0.1 |
| | Optical Flow Estimator | Raft [53] |
| Inference | DDIM Sampling Steps | 50 |

## 9.2 Pliot Study Details

To test the sensitivity of the motion encoder to parts of speech (POS) representing content and motion information, we generated a set of prompts following the template: A [ADJ][NOUN$_1$][VERB][ADV][ADP] the [NOUN$_2$]. We then grouped these prompts according to their respective POS categories. Next, we calculated the pairwise sentence similarity within each group using the "[eot]" token to determine sentence similarity. The average similarity within each group, as well as across different groups, was reported. This setup groups different words with the same POS under the same context, thereby eliminating potential biases introduced by the context.

Table 8: Training dataset of current T2V models.

| Model | Base Model | Training Dataset |
|---|---|---|
| MagicVideo [74] | LDM [41] | WebVid-10M [1] + 10M from HD-VILA-100M [67] + Interal 7M |
| Make-A-Video [46] | - | 2.3B from Laion-5B [44] + WebVid-10M [1] + 10M from HD-VILA-100M[67] |
| Video LDM [4] | LDM [41] | RDS for driving/WebVid-10M [1] for T2V |
| ModelScopeT2V [56] | LDM [41] | 2.3B from Laion-5B [44] + WebVid-10M [1] |
| Show-1 [70] | DeepFloyd[4]+ ModelScopeT2V [56] | WebVid-10M [1] |
| LaVie [59] | Stable Diffusion 1.4 [41] | Laion5B [44] + WebVid-10M [1] + Vimeo-25M [59] |
| PyoCo [14] | eDiff-I [2] | 1.2B text-image dataset + 22.5M text-video dataset |
| VideoFactory [58] | LDM [41] | HD-VG-130M [58] + WebVid-10M [1] |
| EMU VIDEO [45] | Emu [10] | 34M licensed text-video dataset |
| SVD [3] | Stable Diffusion 2.1 [41] | LVD-F [3] (152M) + 250K pre-captioned video clips of high visual fidelity. |
| DEMO | ModelScopeT2V [56] | WebVid-10M [1] |

Table 9: Comparison of different evaluation protocol on MSR-VTT.

| Model | FID ($\downarrow$) | FID-CLIP ($\downarrow$) | FVD ($\downarrow$) | CLIPSIM ($\uparrow$) | Evaluation Protocol |
|---|---|---|---|---|---|
| MagicVideo [74] | 36.50 | - | 1290 | - | Text prompt on test set; unknown number. |
| Make-A-Video [46] | - | 13.17 | - | 0.3049 | FID and CLIPSIM are evaluated on 59794 videos with text prompt from test set. |
| Show-1 [70] | - | 13.08 | 538 | 0.3072 | FID and FVD are evaluated with 2048 videos generated on test set. CLIPSIM is evaluate on 59794 videos with prompts. |
| Video LDM [4] | - | - | - | 0.2929 | CLIPSIM is calculate on 2990 videos with prompts from test set. |
| LaVie [59] | - | - | - | 0.2949 | CLIPSIM is calculate on 2990 videos with prompts from test set. |
| PYoCo [14] | 25.39-22.14 | 10.21-9.73 | - | - | The same as Make-A-Video. |
| VideoFactory [58] | - | - | - | 0.3005 | CLIPSIM is calculate on 2990 videos with prompts from test set. |
| ModelScopeT2V [56] | | 14.89 | 557 | 0.2941 | FID and FVD are evaluated with 2048 videos generated on test set. CLIPSIM is evaluate on 59794 videos with prompts. |
| ModelScopeT2V fine-tuned | | 13.80 | 536 | 0.2932 | Same as ModelScopeT2V |
| **DEMO** | | **11.77** | **422** | **0.2965** | Same as ModelScopeT2V |

We define the sensitivity of our motion encoder as one minus this similarity. The full set of different words within each POS category is defined as follows:

$$ADJ = \{\text{"big", "small", "tall", "short", "fat", "thin", "young", "old"}\}$$
$$NOUN_1 = \{\text{"cat", "dog", "horse", "child", "man", "woman", "bird", "fish"}\}$$
$$VERB = \{\text{"walk", "run", "jump", "crawl", "eat", "swim", "fly", "climb"}\}$$
$$ADV = \{\text{"quickly", "slowly", "suddenly", "steadily", "cautiously",}$$
$$\text{"briskly", "gracefully", "clumsily"}\}$$
$$ADP = \{\text{"across", "over", "through", "beside", "against", "under", "above", "near"}\}$$
$$NOUN_2 = \{\text{"river", "bridge", "mountain", "tree", "house", "lake", "field", "forest"}\}$$

Given these six categories with eight words each, we have a total of $8^6 = 262144$ prompts. It is noteworthy that we did not observe significant differences when using different templates or different sets of words within each POS. The results were consistent across different setups, and we selected these prompts to try to make these prompts meaningful.

## 9.3 Detailed Training and Evaluation for T2V Models

As shown in Table 8, existing T2V models are trained using diverse datasets and strategies, leading to various evaluation standards across different datasets, as detailed in Table 9 and Table 10. Here,

---

[4]https://github.com/deep-floyd/IF

Table 10: Comparison of different evaluation protocols on UCF-101.

| Model | IS ($\uparrow$) | FVD ($\downarrow$) | Evaluation Protocol |
|---|---|---|---|
| MagicVideo [74] | - | 655.00 | Evaluated on videos generated with class labels; unknown number. |
| Make-A-Video [46] | 33.00 | 367.23 | One template sentence per class label; 100 videos per class. |
| Show-1 [70] | 35.42 | 394.46 | One template sentence per class label; 20 videos per prompt for IS; FVD on 2048 sampled videos. |
| Video LDM [4] | 33.45 | 550.61 | Class label only; 100 videos per class. |
| LaVie [59] | - | 526.30 | Class label only; 100 videos per class. |
| PYoCo [14] | 47.76 | 355.19 | One template sentence per class label; 20 videos per prompt for IS; FVD on 2048 sampled videos. |
| VideoFactory [58] | - | 410.00 | One template sentence per class label; 100 videos per class. |
| EMU VIDEO [45] | 42.70 | 606.20 | One template sentence per class label; 100 videos per class. |
| SVD [3] | - | 242.02 | FVD on 13,320 videos using class labels only. |
| ModelScopeT2V [56] | **37.49** | 630.23 | 100 videos per class using class labels only. |
| ModelScopeT2V fine-tuned | 37.21 | 612.53 | 100 videos per class using class labels only. |
| **DEMO** | 36.35 | **547.31** | 100 videos per class using class labels only. |

we detail and justify our evaluation standards. For our evaluation on MSR-VTT, we follow the base model's approach to compute the CLIPSIM on the entire MSR-VTT dataset. For FID computation, CLIP-ViT/B 32 is used to extract the frame features. For FID and FVD, we randomly sample 2048 videos, following the ModelScopeT2V paper's methodology. For our evaluation on UCF-101, to eliminate bias introduced by template sentences (as done in several previous works), we directly use the class labels to compute the IS and FVD scores.

## 9.4 Extended Quantitative Evaluations

To evaluate the generalization of our methods, we applied DEMO on ZeroScope, we report the performance as follows:

Table 11: Quantitative results on ZeroScope.

| Benchmark | Metric | ZeroScope | DEMO+ZeroScope |
|---|---|---|---|
| MSRVTT | FID ($\downarrow$) | 14.57 | 13.59 |
| | FVD ($\downarrow$) | 812 | 543 |
| | CLIPSIM ($\uparrow$) | 0.2833 | 0.2945 |
| UCF-101 | IS ($\uparrow$) | 37.22 | 37.01 |
| | FVD ($\downarrow$) | 744 | 601 |
| WebVid-10M | FID ($\downarrow$) | 11.34 | 10.03 |
| | FVD ($\downarrow$) | 615 | 479 |
| | CLIPSIM ($\uparrow$) | 0.2846 | 0.2903 |
| EvalCrafter | VQA_A ($\uparrow$) | 27.76 | 33.02 |
| | VQA_T ($\uparrow$) | 33.87 | 37.28 |
| | IS ($\uparrow$) | 14.20 | 15.28 |
| | ActionScore ($\uparrow$) | 67.78 | 72.55 |
| | MotionAC-Score ($\uparrow$) | 44 | 62 |
| | FlowScore ($\uparrow$) | 1.10 | 5.25 |
| Vbench | MotionDynamics ($\uparrow$) | 42.72 | 70.28 |
| | HumanAction ($\uparrow$) | 67.36 | 88.34 |
| | TemporalFlickering ($\downarrow$) | 97.39 | 94.83 |
| | MotionSmoothness ($\uparrow$) | 97.92 | 95.72 |

## 9.5 Extended Qualitative Evaluations

In this section, we provide extended qualitative comparison between our method and the baseline.

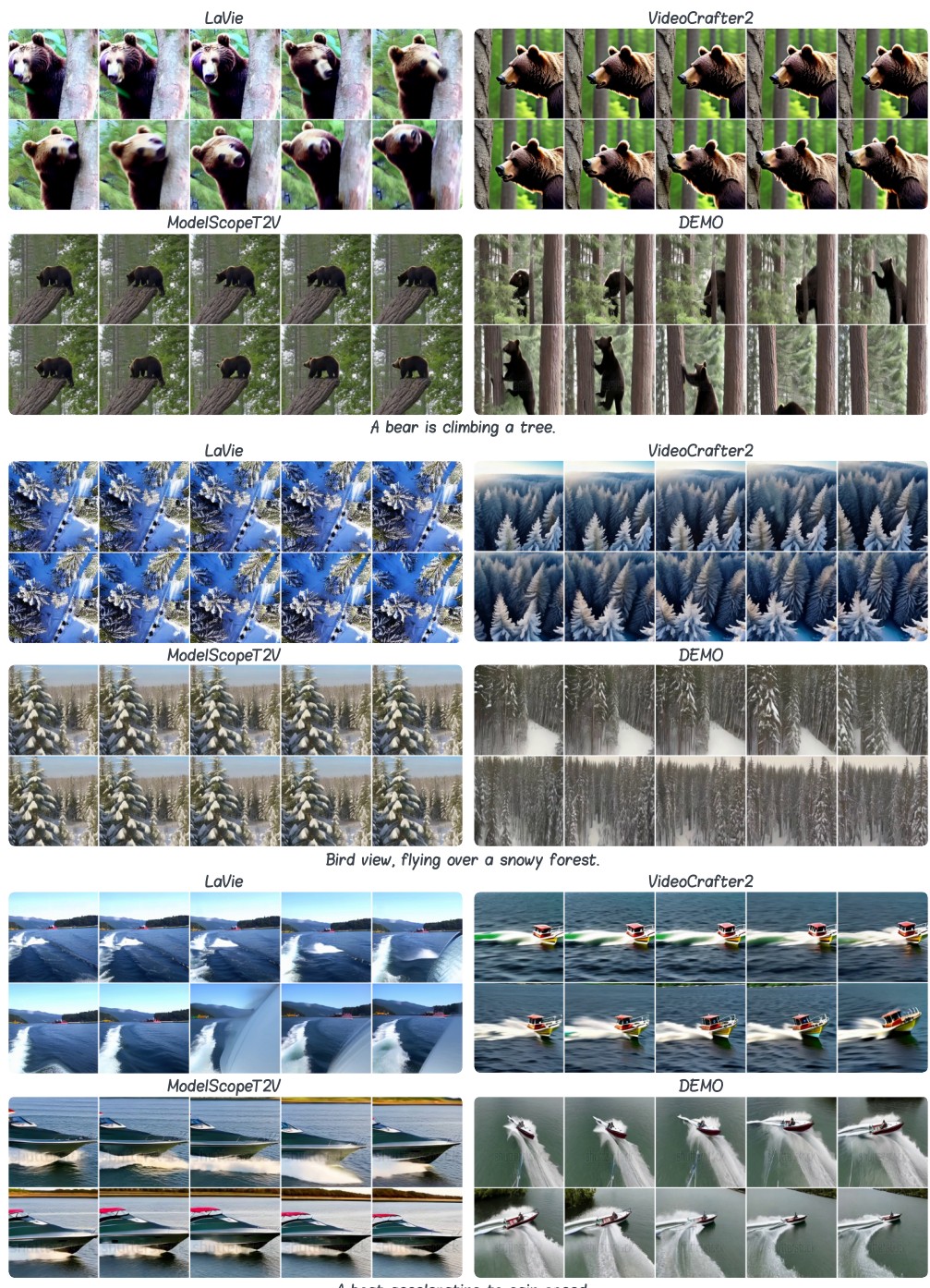

Figure 5: **Extended qualitative comparison.** Each video is generated with 16 frames. We display frames 1, 2, 4, 6, 8, 10, 12, 14, 15, and 16, arranged in two rows from left to right. Full videos are available in the supplementary materials.

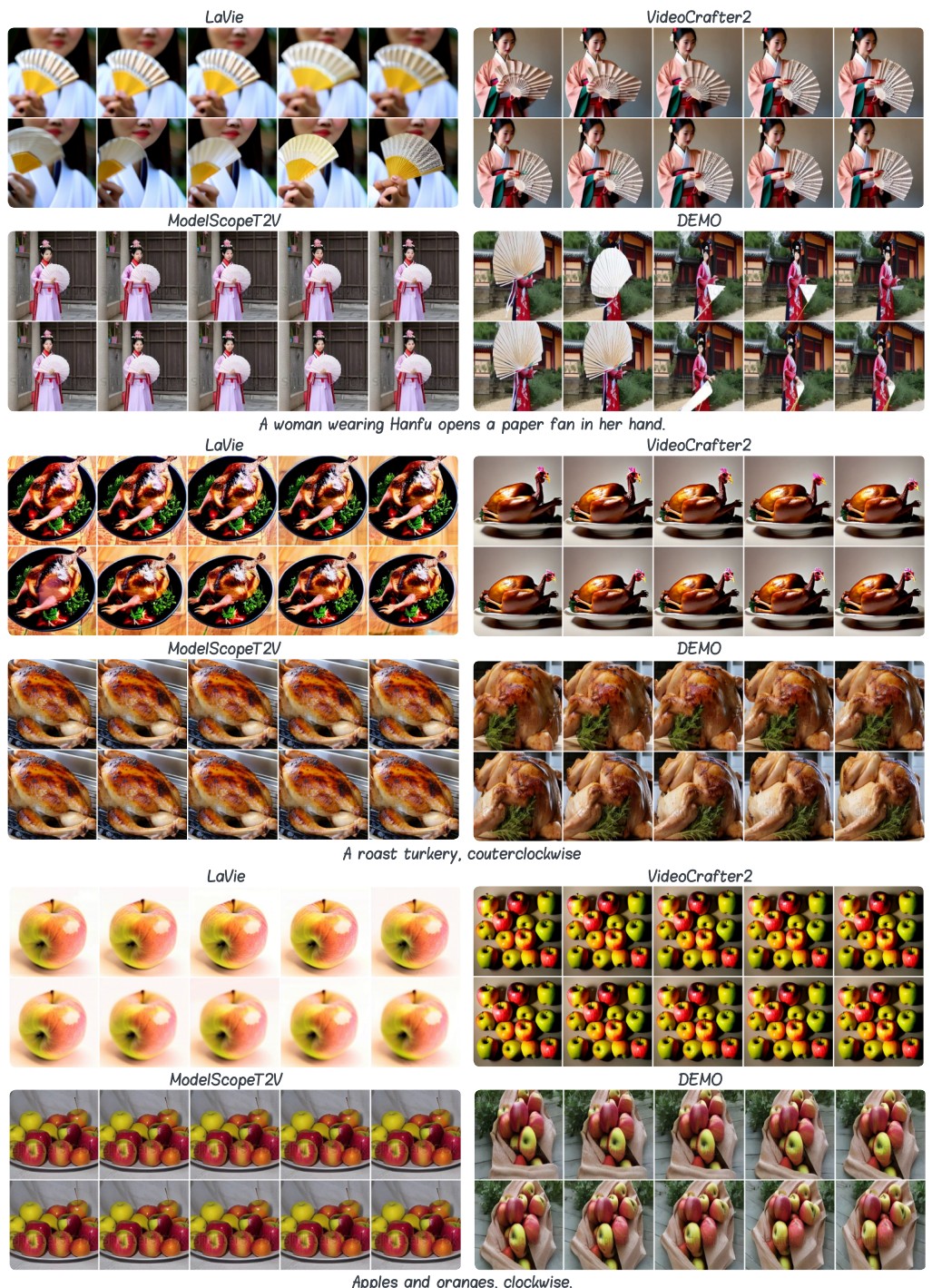

Figure 6: **Extended qualitative comparison.** Each video is generated with 16 frames. We display frames 1, 2, 4, 6, 8, 10, 12, 14, 15, and 16, arranged in two rows from left to right. Full videos are available in the supplementary materials.

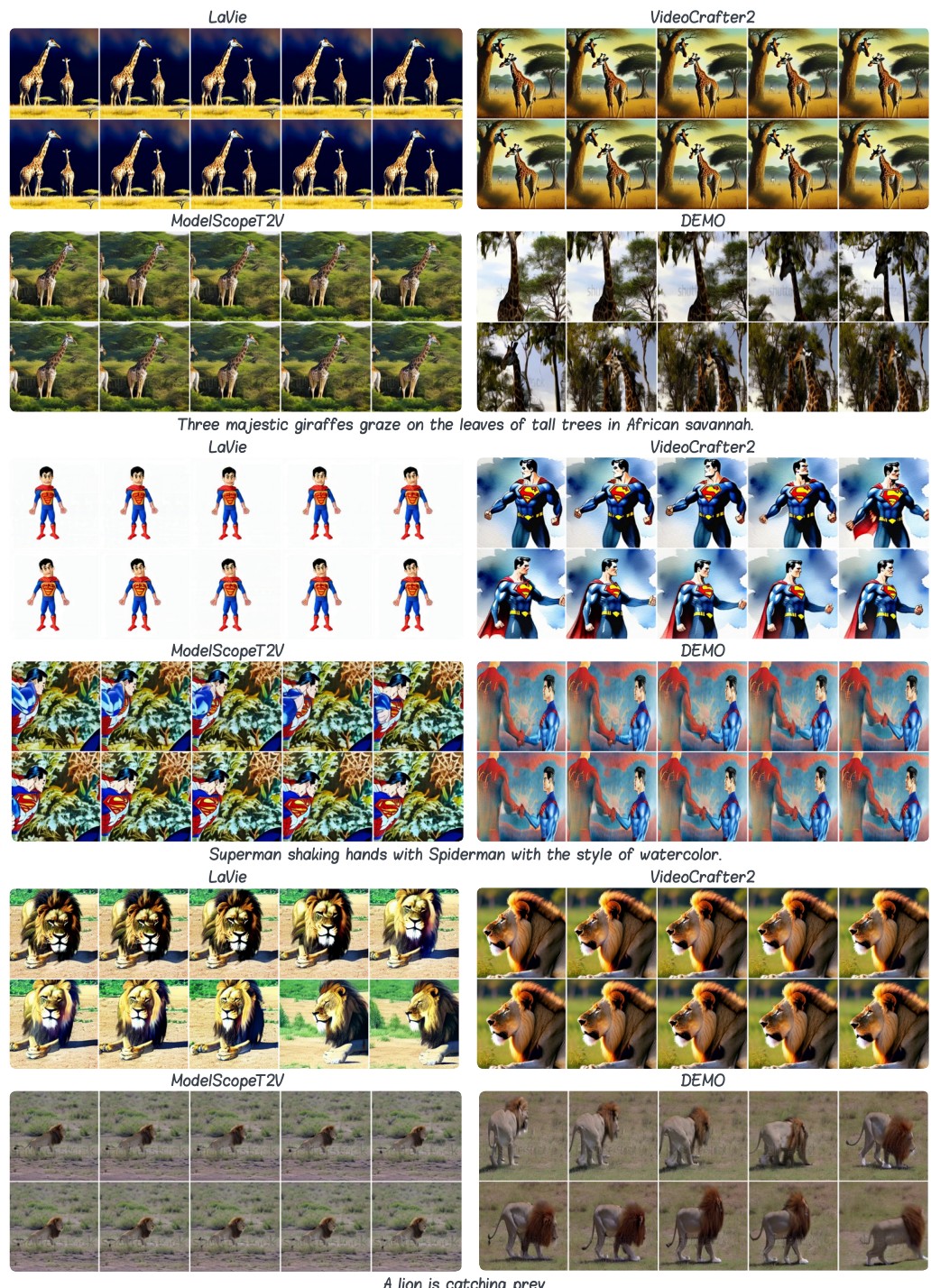

Figure 7: **Extended qualitative comparison.** Each video is generated with 16 frames. We display frames 1, 2, 4, 6, 8, 10, 12, 14, 15, and 16, arranged in two rows from left to right. Full videos are available in the supplementary materials.

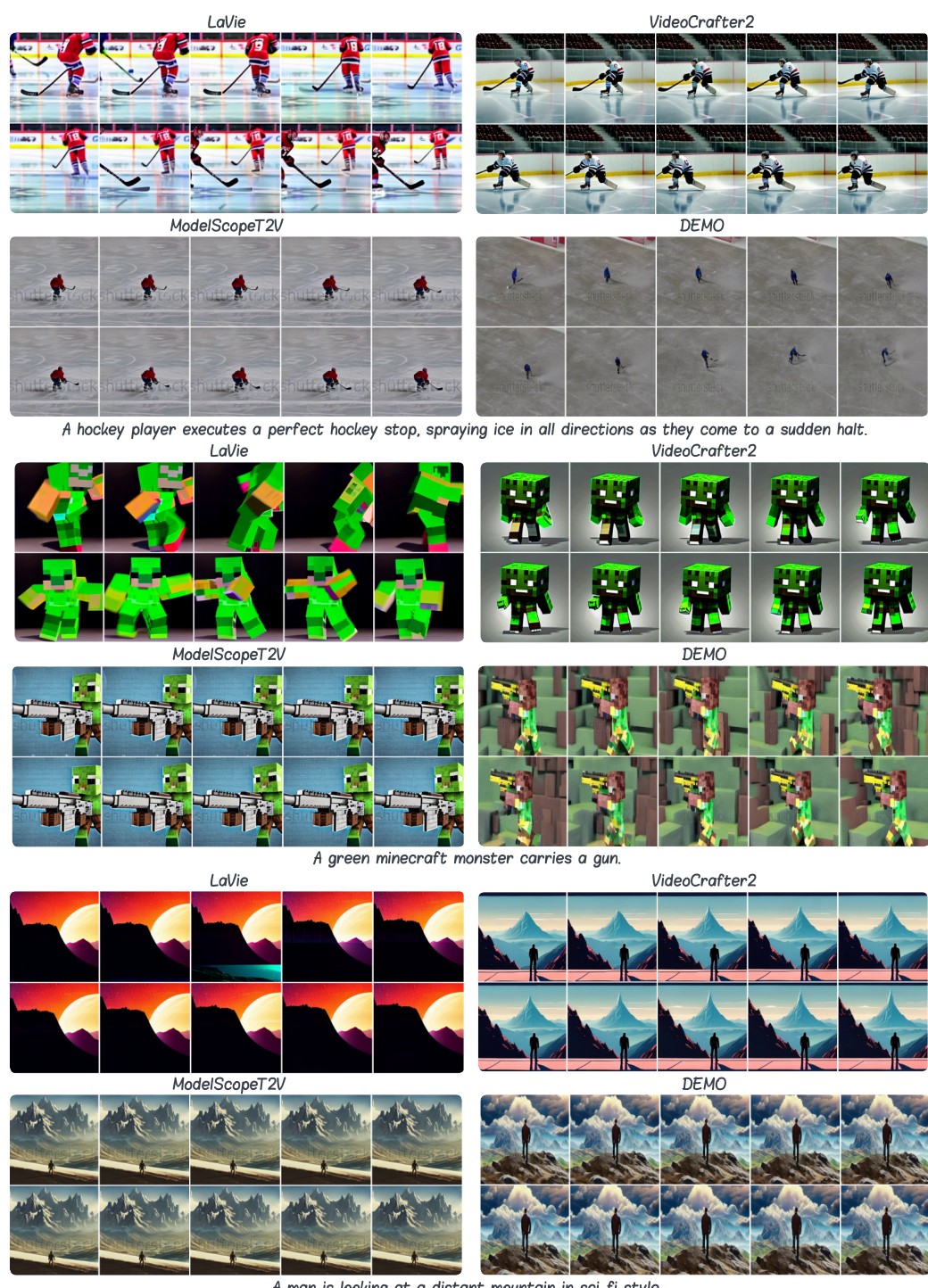

Figure 8: **Extended qualitative comparison.** Each video is generated with 16 frames. We display frames 1, 2, 4, 6, 8, 10, 12, 14, 15, and 16, arranged in two rows from left to right. Full videos are available in the supplementary materials.

## 9.6 Human Evaluation

To further assess the qualitative performance of our proposed method, we conducted a user study comparing our approach (DEMO) with several state-of-the-art video generation models. We randomly

selected 50 prompts from EvalCrafter [31], ensuring diversity across scenes, styles, and objects. For each comparison, 15 annotators evaluated the generated videos in terms of three main criteria: text-video alignment, visual quality, and motion quality. The study compared our method with ModelScopeT2V, LaVie, and VideoCrafter2.

The participants were asked to select their preferred video between the two models for each prompt. The comparative results are summarized in Table 12. Specifically, DEMO consistently outperformed ModelScopeT2V, LaVie, and VideoCrafter2, particularly in terms of motion quality, where it achieved a preference rate of 74% over ModelScopeT2V. Additionally, DEMO was favored in text-video alignment and visual quality by 62% and 66%, respectively. However, when compared to LaVie and VideoCrafter2, DEMO showed a lower performance in visual quality, which can be attributed to differences in training datasets. LaVie and VideoCrafter2 use higher-quality video and image datasets, such as Vimeo-25M [59] and JDB [52], respectively, while DEMO and ModelScopeT2V are trained on the WebVid10M dataset, which is lower in visual quality.

Furthermore, we conducted an additional user study to evaluate the effectiveness of our proposed video-motion supervision term, $\mathcal{L}_{\text{video-motion}}$. The results indicated that our method with motion supervision outperformed the version without motion supervision, achieving win rates of 58%, 56%, and 72% in text-video alignment, visual quality, and motion quality, respectively. These findings highlight the significant improvements brought by the video-motion supervision in generating smoother and more realistic motion dynamics.

Table 12: User Study Results: Comparison between DEMO and Other Models

| Methods | Text-Video Alignment | Visual Quality | Motion Quality |
|---|---|---|---|
| DEMO vs ModelScopeT2V | 62% | 66% | 74% |
| DEMO vs LaVie | 56% | 46% | 62% |
| DEMO vs VideoCrafter2 | 60% | 42% | 52% |
| DEMO vs DEMO w/o $\mathcal{L}_{\text{video-motion}}$ | 58% | 56% | 72% |

In summary, the user study provides strong evidence that DEMO improves motion quality without sacrificing text-video alignment. Despite the lower visual quality compared to models trained on high-quality datasets, our method demonstrates its strength in motion generation, a key aspect of video realism.

