# OpenReview forum: "Enhancing Motion in Text-to-Video Generation with Decomposed Encoding and Conditioning"
_NeurIPS.cc/2024/Conference — NeurIPS 2024 poster_

### Official Review · Reviewer_5zwJ · 2024-06-14

**Soundness:** 3
**Presentation:** 3
**Contribution:** 3
**Rating:** 7
**Confidence:** 3

**Summary:**

This paper introduces DEMO, a text to video diffusion approach that aims at enhancing the movement in the video produced by text to video models. In additional to the classical diffusion loss, DEMO introduces:

* __A text motion loss__: the CLIP text encoder used for the temporal cross attention is fine-tuned - the loss matches the text motion to the motion of the video as computed with the optical flow
* __A regularization__: to avoid the CLIP text encoder used for the temporal cross attention to diverge too much from the CLIP encoder used for the spatial cross attention
* __A video motion loss__: to encourage the frame latents output by model to follow the difference between frames in the actual video

The model is trained on WebVid and compared to other models trained on WebVid on both video quality and motion quality.

**Strengths:**

Technical contribution is strong:
* The need for motion focused text encoder is well argued
* The experiments show that this addition contributes into increasing movement

Technical clarity is also good:
* The different losses are presented in a clear and detailed way.

**Weaknesses:**

The comparison with other approaches is not 100% fair in the sense that additional parameters are being trained on the text encoder side (a CLIP encoder).

It might also be that additional parameters are being trained on the temporal cross attention side. This is actually unclear to me, because at the beginning of the method section (section 3) the author define the U-Net 3D this way: "the temporal transformer consists of temporal self-attention and feed-forward layers", while some papers do use cross-attention in the temporal self-attention as well.

**Questions:**

The author define the U-Net 3D this way: "the temporal transformer consists of temporal self-attention and feed-forward layers", while some papers do use cross-attention in the temporal self-attention as well. Is that a mistake in the text, or to which specific papers are you referring to as the "vanilla" 3D U-Net ?

Is the temporal attention similar to Make-A-Video and is operating on a single patch ? Or do you use full attention across all patches of all other frames ? That also has an impact on the fairness of comparisons / ablations of the importance of the losses that you introduce.

---

> ### Author Rebuttal · Authors · 2024-08-06
>
> Thank you for your detailed feedback and insightful questions. Below are our responses to the concerns raised:
>
> **Q1**: The comparison with other approaches is not 100\% fair since additional parameters (a CLIP encoder) are trained.
>
> **Response**: We appreciate your concern about the fairness of comparisons due to additional parameters being trained on the text encoder side (a CLIP encoder). We conducted an experiment to rule out the effect of the additional parameters in the text encoder. Specially, we trained the baseline ModelScopeT2V model with a new CLIP text encoder while keeping its original training loss. We then compared its performance to the model without the additional CLIP encoder parameters (ModelScopeT2V fine-tuned column in the table). From the experimental results, we observed that the performance of the baseline model with the additional text encoder parameters is similar to that of directly fine-tuning the ModelScopeT2V without additional parameters. This indicates that without appropriate supervision, the impact of the additional parameters in the text encoder has only a marginal influence on the overall performance.
>
> |Benchmark|Metric|ModelScopeT2V|ModelScopeT2V fine-tuned|ModelScopeT2V + text encoder|DEMO|
> |---------|------|-------------|-----------------------|----------------------------|----|
> |MSRVTT|FID (↓)|14.89|13.80|13.98|11.77|
> ||FVD (↓)|557|536|552|422|
> ||CLIPSIM (↑)|0.2941|0.2932|0.2935|0.2965|
> |UCF-101|IS (↑)|37.55|37.21|37.66|36.35|
> ||FVD (↓)|628.17|612.53|601.25|547.31|
> |WebVid-10M|FID (↓)|11.14|10.53|10.45|9.86|
> ||FVD (↓)|508|461|458|351|
> ||CLIPSIM (↑)|0.2986|0.2952|0.2967|0.3083|
> |EvalCrafter|VQA_A (↑)|15.12|15.89|16.21|19.28|
> ||VQA_T (↑)|16.88|16.39|16.34|15.65|
> ||IS (↑)|14.60|14.92|15.02|17.57|
> ||Action Score (↑)|75.88|74.23|75.20|78.22|
> ||Motion AC-Score (↑)|44|40|46|58|
> ||Flow Score (↑)|2.51|2.72|2.44|4.89|
> |Vbench|Motion Dynamics (↑)|62.50|63.75|63.50|68.90|
> ||Human Action (↑)|90.40|90.40|90.20|90.60|
> ||Temporal Flickering (↑)|96.02|96.35|95.45|94.63|
> ||Motion Smoothness (↑)|96.19|96.38|96.22|96.09|
>
>
> **Q2**: The definition of the 3D U-Net in your paper states: "the temporal transformer consists of temporal self-attention and feed-forward layers." Some papers also use cross-attention in temporal self-attention. Is this a mistake, and which specific papers are you referring to as the "vanilla" 3D U-Net?
>
> **Response**:
> Thank you for pointing out the ambiguity in our definition of the 3D U-Net. In the methodology section, when we refer to the 3D U-Net, we are specifically referencing the ModelScopeT2V model. This model employs spatial transformers for spatial conditioning and temporal transformers with temporal self-attention for smoothing individual frames along the temporal dimension. We will clarify this in the revised manuscript to avoid further confusion.
>
>
> **Q3**: How your temporal attention is performed? Do you perform the attention on a single patch or full attention across all patches of all other frames?
>
> **Response**:
> Thank you for raising your question about the attention mechanism in our model. The temporal cross-attention in our model operates on a single patch across the frame dimension rather than full attention across all patches of all frames.
>
> Our approach involves decomposing the full 3D cross-attention into two separate components: a 2D spatial cross-attention (content conditioning) and a 1D temporal cross-attention (motion conditioning). This design choice enables us to: 1) Preserve the text-to-video generation capability (generate 2D content for each individual frame separately) of the base model. 2) Allow the text instructions to control the temporal dynamics of the video effectively. By separating the spatial and temporal attention mechanisms, we achieve a balance between computational efficiency and performance. This approach ensures fair comparisons and meaningful ablations of the introduced losses, demonstrating the effectiveness of our conditioning mechanism.

---

> > ### Comment · Reviewer_5zwJ · 2024-08-12
> > **Answer to Rebuttal**
> >
> > Thanks a lot to the authors for answering my concerns. I will keep my rating the same.

---

> > > ### Author Response · Authors · 2024-08-13
> > >
> > > Dear Reviewer,
> > >
> > > We sincerely appreciate your thoughtful comments, recognition of our response, and support for our work.
> > >
> > > Sincerely,
> > >
> > > The Authors

---

### Official Review · Reviewer_WCt2 · 2024-07-12

**Soundness:** 3
**Presentation:** 3
**Contribution:** 3
**Rating:** 7
**Confidence:** 4

**Summary:**

This paper introduces novel framework called DEcomposed MOtion (DEMO), which enhances motion synthesis in T2V generation by decomposing both text encoding and conditioning into content and motion components. Authors investigate sensitivity of CLIP text encoder to motion descriptions and propose to condition the model on content and motion separately. To obtain a motion encoder, authors propose fine-tune separate CLIP text encoder with losses that  encourage its [eot] token to describe motion better. To this end, authors introduce novel text-motion and video-motion losses to encourage the motion conditioning module to generate and render motion dynamics. Experiment results show that proposed method enhances motion of the T2V model.

**Strengths:**

1. Importance. The proposed method successfully tackles the an important problem of enhancing motion quality in videos generated by T2V models.

2. Results. Experiment results demonstrate the effectiveness of the proposed method.

3. Clarity. The text of the paper is well written and  easy to follow.

**Weaknesses:**

1. Lack of user study. Qualitative results are supported only by examples of several videos. Such results are not statistically representative. Please See Question 1.

**Questions:**

1. I strongly recommend that the authors incorporate a user study in their research to obtain qualitative results. Additionally, conducting ablation studies with a larger sample size, consisting of a minimum of 50 distinct prompts, and involving at least 15 independent assessors, would greatly enhance the robustness and reliability of the findings.

2. On line 127, the authors mention calculating optical flow between attention maps of different frames. However, the methodology employed to perform this operation lacks clarity in the paper. It is essential for the authors to provide a more detailed explanation of the procedures used to compute optical flow between attention maps. Furthermore, considering that attention maps in diffusion models are often noisy, it is crucial to address the reliability and potential limitations associated with this computation.

**Limitations:**

The authors adequately addressed the limitations.

---

> ### Author Rebuttal · Authors · 2024-08-06
>
> **Q1**: We strongly recommend to incorporate a user study in the research to obtain statistical qualitative results.
>
> **Response**:
> Thank you for your excellent suggestion. We conducted a user study to compare our method with other video generation models. We selected 50 prompts from EvalCrafter [1], covering diverse scenes, styles, and objects. For each model comparison, 15 annotators were asked to select their preferred video from three options: method 1, method 2, and comparable results. They evaluated the videos based on visual quality, motion quality, and text-video alignment. The results are shown as follows:
> |Methods|Text-Video Alignment|Visual Quality|Motion Quality|
> |-------|---------------------|--------------|--------------|
> |DEMO vs ModelScopeT2V|62%|66%|74%|
> |DEMO vs LaVie|56%|46%|62%|
> |DEMO vs VideoCrafter2|60%|42%|52%|
> |DEMO vs DEMO w/o $\mathcal{L}_{\text{video-motion}}$|58%|56%|72%|
>
> Specifically, 74\% of user feedback indicated that DEMO has better motion quality compared to the baseline ModelScopeT2V. Additionally, 62\% and 66\% of the user feedback agreed that DEMO has better text-video alignment and visual quality, respectively. These findings align with our quantitative results, demonstrating that DEMO significantly improves motion quality without compromising visual quality and text-video alignment.
>
> We also performed a user study to test the effect of our video-motion supervision term, $\mathcal{L}_{\text{video-motion}}$, which is designed to guide the motion conditioning module to generate better motion dynamics. We achieved win rates of 58\%, 56\%, and 72\% in terms of text-video alignment, visual quality, and motion quality, respectively, compared to DEMO without video-motion supervision. Additionally, we observed that both ModelScopeT2V and our improved version, DEMO, have lower visual quality compared to LaVie and VideoCrafter2. We attribute this to the fact that the training dataset for ModelScopeT2V and DEMO (specifically WebVid10M) is low quality visually. In contrast, LaVie uses the high-quality video dataset Vimeo-25M [2], and VideoCrafter2 adopts the high-quality image dataset JDB [3] for better visual quality.
>
> We appreciate the recommendation to incorporate this into the revised paper, and we will include a summary of these findings to enhance the robustness and reliability of our qualitative results.
>
> **Q2**: How to calculate the optical flow between attention maps is not clearly described. Furthermore, considering that attention maps in diffusion models are often noisy, it is crucial to address the reliability and potential limitations associated with this computation.
>
> **Response**: Thank you for pointing out the lack of clarity in our explanation. In our approach, we utilize RAFT [4] to calculate the optical flow, as detailed in Table 6 of the appendix. Basically, we treat cross-attention maps as kind of videos but in a different space compared with real world videos. We then directly apply RAFT to extract the optical flows. We recognize that attention maps in diffusion models can be noisy, which poses challenges for reliable optical flow computation. To address this, we tried a simple thresholding strategy that zeroes out regions with little activation. This approach resulted in slight improvements in the stability and accuracy of the optical flow calculations. Nonetheless, we acknowledge that this is a preliminary solution and that more sophisticated strategies could yield better results.
>
> In the revised paper, we will provide a more detailed explanation of our methodology for calculating optical flow. This will include a step-by-step description of how we preprocess the attention maps, apply the RAFT model, and handle the inherent noise in the attention maps. Additionally, we will discuss the potential limitations of our current approach and outline possible directions for future work to enhance the robustness and reliability of optical flow calculations in noisy conditions.
>
> [1] Teed, Z., & Deng, J. (2020). Raft: Recurrent all-pairs field transforms for optical flow. In Computer Vision–ECCV 2020: 16th European Conference, Glasgow, UK, August 23–28, 2020, Proceedings, Part II 16 (pp. 402-419). Springer International Publishing.
>
> [2] Wang, Y., Chen, X., Ma, X., Zhou, S., Huang, Z., Wang, Y., ... & Liu, Z. (2023). Lavie: High-quality video generation with cascaded latent diffusion models. arXiv preprint arXiv:2309.15103.
>
> [3] Sun, K., Pan, J., Ge, Y., Li, H., Duan, H., Wu, X., ... & Li, H. (2024). Journeydb: A benchmark for generative image understanding. Advances in Neural Information Processing Systems, 36.
>
> [4] Teed, Z., & Deng, J. (2020). Raft: Recurrent all-pairs field transforms for optical flow. In Computer Vision–ECCV 2020: 16th European Conference, Glasgow, UK, August 23–28, 2020, Proceedings, Part II 16 (pp. 402-419). Springer International Publishing.

---

> ### Comment · Reviewer_WCt2 · 2024-08-11
> **Response to Rebuttal**
>
> Dear Authors,
>
> Thank you for your clarifications and conducted user study. My major concerns were addressed, so I raise my score to **7: Accept**.
>
> However, the fact that directly applying RAFT to extract the optical flows from attention maps works that well is surprising to me. Intuitively, it should be an out of domain input for RAFT. Looking forward for your detailed explanation of this part of pipeline in the camera-ready version of the paper.
>
> Best regards,
>
> Reviewer WCt2

---

> > ### Author Response · Authors · 2024-08-12
> >
> > Dear Reviewer,
> >
> > We sincerely appreciate your thoughtful comments, recognition of our response, and support for our work.
> >
> > Sincerely,
> >
> > The Authors

---

### Official Review · Reviewer_gMRr · 2024-07-13

**Soundness:** 3
**Presentation:** 2
**Contribution:** 3
**Rating:** 4
**Confidence:** 3

**Summary:**

This paper proposes a method for enhancing motion synthesis in T2V generation by decomposing both text encoding and conditioning into content and motion components, called Decomposed Motion (DEMO). To address the issue of inadequate motion representation in text encoding, they decompose text encoding into content encoding and motion encoding processes, focusing on spatial information and temporal information respectively. To solve the problem of reliance on spatial-only text conditioning, they also decompose the text conditioning process into content and motion dimensions to integrate spatial and temporal conditions. Additionally, they incorporate text-motion and video-motion supervision to enhance the model’s understanding and generation of motion.

**Strengths:**

1. The paper is well written overall and it is easy to follow along with the presented concepts and results.
2. The paper includes experimental results that validate the authors' claims, as well as ablations that offer a better understanding of model design choices.
3. The motivation for proposing the method is well demonstrated experimentally.
4. The proposed method is efficient in training, has a small number of parameters and a low inference burden, and does not require additional reference information.

**Weaknesses:**

1. There is too little content in the related work section.
2. There is a lack of contrast experiments with other improved methods that rely on external references to exhibit performance differences.
3. The proposed method is generalized, so why not validate it on other BASE models? From qualitative and quantitative experiments, the performance and various indexes of ModelScope are poor, and while DEMO is useful, it is unclear if there is any improvement when adding DEMO to models with better performance. There is a lack of relevant experimental evidence.
4. The provided videos are less impressive. The separate files of other methods make it difficult for reviewers to identify the advantages of the proposed method, it would be better to make the comparison in a single file.

**Questions:**

In line 196, “showing marked improvement in individual frame quality compared to the baseline,” can you explain why the quality of a single frame is increased? As I understand, DEMO is designed to increase the model's motion generation capability.

**Limitations:**

See weakness.

---

> ### Author Rebuttal · Authors · 2024-08-06
>
> Thank you for your detailed feedback. Below are our responses to the concerns raised:
>
> **Q1**: The related work section contains too little content.
>
> **Response**: We acknowledge that the related work section is limited due to page constraints. In the revised paper, we will expand this section and include additional relevant studies (references omitted here due to limits):
>
> **T2V Generation**: T2V has advanced significantly, leveraging T2I success. The first T2V model, VDM, introduced a space-time factorized U-Net for temporal modeling, training on both images and videos. For high-definition videos, models like ImagenVideo, Make-A-Video, LaVie, and Show-1 use cascades of diffusion models with spatial and temporal super-resolution. MagicVideo, Video LDM, and LVDM apply latent diffusion for video, working in a compressed latent space. VideoFusion separates video noise into base and residual components. ModelScopeT2V, the first open-source T2V diffusion model, uses 1D convolutions and attention to approximate 3D operations. Stable Video Diffusion (SVD) divides the process into T2I pre-training, video pre-training, and fine-tuning.
>
> **Q2**: There is a lack of contrast experiments with other methods that rely on external references.
>
> **Response**: Thank you for raising your concern. Here we list the key differences between our method and other methods that rely on external references to address your concern.
>
> 1) Different Goal: Our focus is to enhance motion synthesis in general text-to-video generation without relying on external references, which are often infeasible in real-world scenarios. Methods using external references, like video customization or controllable video generation, aim to enable more control to generate videos with specific object appearances, motions, and other details, which is not our goal.
>
> 2) Different Information Sources: Our model takes only sparse textual descriptions. It cannot directly take external references without additional training and adaptation, which is beyond the scope of our paper. Methods using external references have access to more detailed visual signals. Comparing our approach with such methods is unfair due to the disparity in information density.
>
> **Q3**: There is a lack of experiments with other base model to show the generalization of the proposed method. Additionally, the performance of the baseline model is poor.
>
> **Response**:
> We agree that demonstrating the generalization of our method on other base models would be beneficial. To this end, we have applied DEMO to ZeroScope [1] and included the experimental results in the table below. The results are consistent with those obtained using ModelScopeT2V, showing significant improvements in motion quality without loss of visual quality.
>
> |Benchmark|Metric|ZeroScope|DEMO+ZeroScope|
> |---------|------|---------|--------------|
> |MSRVTT|FID(↓)|14.57|13.59|
> ||FVD(↓)|812|543|
> ||CLIPSIM(↑)|0.2833|0.2945|
> |UCF-101|IS(↑)|37.22|37.01|
> ||FVD(↓)|744|601|
> |WebVid-10M|FID(↓)|11.34|10.03|
> ||FVD(↓)|615|479|
> ||CLIPSIM(↑)|0.2846|0.2903|
> |EvalCrafter|VQA_A(↑)|27.76|33.02|
> ||VQA_T(↑)|33.87|37.28|
> ||IS(↑)|14.20|15.28|
> ||ActionScore(↑)|67.78|72.55|
> ||MotionAC-Score(↑)|44|62|
> ||FlowScore(↑)|1.10|5.25|
> |Vbench|MotionDynamics(↑)|42.72|70.28|
> ||HumanAction(↑)|67.36|88.34|
> ||TemporalFlickering(↑)|97.39|94.83|
> ||MotionSmoothness(↑)|97.92|95.72|
>
> Additionally, ModelScopeT2V is the only fully open-source text-to-video model (dataset, model, and code). While its performance may be lower than some closed-source models, it performs reasonably well especially in motion quality. Below are its performance rankings across various benchmarks against both closed-source and open-source models:
>
> |Benchmark|Metric|Ranking|
> |---|---|---|
> |DEVIL [2]|Dynamics Quality|3/10|
> ||Dynamics Control|5/10|
> |FETV [3]|Temporal Quality|1/4|
> |T2V-CompBench [4]|Motion|4/20|
> |VBench [5]|Dynamic Degree|6/29|
> ||Human Action|13/29|
> ||Temporal Style|10/29|
>
>
> **Q4**: The provided videos are less impressive and the separate files make it difficult to compare those methods.
>
> **Response**: Please refer to Q1 in Global Response.
>
> **Q5**: Why DEMO can improve the individual frame quality (FID) since DEMO is designed to enhance the motion generation?
>
> **Response**: Thank you for raising this concern. The confusion arises from the fact that individual frame quality (FID) involves both the realism of the frame itself and the diversity of frames. The FID [6] calculate the Fréchet Distance between generated frames and real frames as follows:
> \begin{equation}
>     d(P_R, P_G) = \| \mu_R - \mu_G \|^2 + \text{Tr}(\Sigma_R + \Sigma_G - 2(\Sigma_R \Sigma_G)^{\frac{1}{2}})
> \end{equation}
> where $\mu$, and $\Sigma$ are mean, covariance of real ($R$) and generated ($G$) frames. The mean term measures frame realism by comparing the average feature representation of generated and real frames. The covariance term assesses frame diversity by capturing how well generated frames replicate the variability and feature relationships of real frames.
>
> Our base model uses content conditioning for each individual frame and temporal self-attention for smoothing those frames, which improves motion smoothness but limits motion dynamics and frame diversity. In contrast, DEMO introduces a separate motion conditioning module, enhancing motion dynamics and generating more diverse frames. This increased diversity leads to a better FID score, thus improving individual frame quality.
>
> [1] Zeroscope. https://huggingface.co/cerspense/zeroscope_v1_320s. 2023.
>
> [2] Evaluation of Text-to-Video Generation Models: A Dynamics Perspective. 2024.
>
> [3] Fetv: A benchmark for fine-grained evaluation of open-domain text-to-video generation. 2024.
>
> [4] T2V-CompBench: A Comprehensive Benchmark for Compositional Text-to-video Generation. 2024.
>
> [5] Vbench: Comprehensive benchmark suite for video generative models. 2024.
>
> [6] Gans trained by a two time-scale update rule converge to a local nash equilibrium. 2017.

---

### Official Review · Reviewer_uuEr · 2024-07-13

**Soundness:** 3
**Presentation:** 3
**Contribution:** 2
**Rating:** 5
**Confidence:** 5

**Summary:**

This paper aims to improve the motion dynamics generation of the text-to-video generation models. It proposes a framework to decompose both text encoding and conditioning into content and motion components. For text encoding, a CLIP encoder is fine-tuned to encode the motion information in the text prompts better. For text conditioning,  it introduces a motion conditioning module to incorporate motion information in the denoising process.

**Strengths:**

1. The general idea is interesting, i.e. decompose the text encoding and conditioning into content and motion components to capture the motion prompts and generate motion dynamics better.
2. The pilot study is interesting, which investigates that the CLIP encoder tends to be less sensitive to the motion instructions in the text prompts.
3. This paper is technically clear and easy to follow.

**Weaknesses:**

1. For the pilot study in Fig. 1, only one kind of sentence template is applied. Will the drawn conclusion be the same when the text prompts changes to other formats? This concern is not addressed in the paper.
2. In Fig.3, it is hard to tell which rows are generated by the base model and which rows are generated by the fine-tuned model.
3. From line 179 to line 183, it is hard to evaluate the correctness of the conclusion. Because it is quite hard to distinguish the two generated videos shown in Fig. 3. They look almost the same.
4. For qualitative evaluations. The improvements in generated motions in most shown cases are limited.

**Questions:**

Please refer to the weaknesses.

**Limitations:**

The limitations are discussed in the paper.

---

> ### Author Rebuttal · Authors · 2024-08-06
>
> We appreciate your valuable feedback. Below are our responses to the concerns raised:
>
> **Q1**: In the pilot study, employing only one kind of sentence template may not be sufficient to draw the conclusion that the text encoder is less sensitive to the motion instructions in the text prompt.
>
> **Response**: We thank the reviewer for highlighting this concern. Due to page limitations, we included the details of our pilot study in the appendix. As stated in the appendix (lines 527 to 530), "It is noteworthy that we did not observe significant differences when using different templates or different sets of words within each POS. The results were consistent across different setups, and we selected these prompts to try to make these prompts meaningful." We will revise the caption for our pilot study (Fig.1) to include this information and to address this concern as follows:
>
> "We generated a set of prompts (262144 in total) following a fixed
> template, grouping them according to the different parts of speech (POS). These grouped texts are
> then passed into the CLIP text encoder, and we calculate the sensitivity as the average sentence
> distance within each group. As shown on the left-hand side, compared to POS representing content,
> CLIP is less sensitive to POS representing motion. (Results are consistent across different templates and different sets of words within each POS. Further details can be found in the appendix.)"
>
> **Q2**: In Fig.3, it is hard to tell which rows are generated by the base model and which rows are generated by the fine-tuned model.
>
> **Response**: Thank you for pointing out the potential confusion caused by the arrangement of video frames in Fig.3. This complexity arises because our quantitative evaluation involves multiple dimensions. We need to compare our model with other models, show the differences within each video, and evaluate multiple text inputs.
>
> In Fig.3, we present video frames generated by four different models: LaVie, VideoCrafter2, ModelScopeT2V (base model), and DEMO (fine-tuned model), across three different text prompts. As mentioned in the caption of Fig.3, "Each video is generated with 16 frames. We display frames 1, 2, 4, 6, 8, 10, 12, 14, 15, and 16, arranged in two rows from left to right. Full videos are available in the supplementary materials."
>
> To improve clarity, we plan to display these videos in a GIF format in our revised paper. This will help illustrate the differences more effectively. Please refer to Fig.1 in the PDF of Global Response.
>
>
> **Q3**: It is hard to evaluate that our DEMO significantly outperforms the base model with the text prompt "Slow motion flower petals fall from a blossom, landing softly on the ground (the first example in Fig.3)" as the frames look almost the same.
>
> **Response**: Thank you for pointing out the potential confusion in our presentation of video frames. The motion between different frames, especially those involving many objects such as flower petals, can be difficult to observe with static frames alone. We have provided video files in the supplementary materials, which clearly show significant differences between these two videos.
>
> Additionally, we plan to revise Fig.3 in our paper to better highlight these differences. Please refer to Q1 and Fig.1 in the Global Response.
>
>
> **Q4**: For qualitative evaluations, the improvements in generated motions in most shown cases are limited.
>
> **Response**: Thanks for raising your concern. This is due to the fact that the improvements of motion are hardly to observe by only showing sequence of static frames. We have provided video files in the supplementary materials to better observe these improvements. Additionally, we plan to revise Fig.3 in our paper to better highlight these differences. Please refer to Q1 and Fig.1 in the Global Response.

---

> > ### Comment · Reviewer_uuEr · 2024-08-12
> > **Thanks for the rebuttal.**
> >
> > The authors have addressed most of my concerns. I decide to keep my initial positive rating.

---

> > > ### Author Response · Authors · 2024-08-12
> > >
> > > Dear Reviewer,
> > >
> > > We sincerely appreciate your thoughtful comments, recognition of our response, and support for our work.
> > >
> > > Sincerely,
> > >
> > > The Authors

---

### Author Rebuttal · Authors · 2024-08-06

**Q1**:The presentation of Fig.3 (qualitative results) is not good. The improved motion are hardly to observe and it is difficult to compare DEMO with other methods.

**Response**: We thank all the reviewers for highlighting this concern. Qualitative comparison of videos generated by different models is inherently challenging, especially when focusing on the dynamic aspects of video generation. We acknowledge that displaying only sequence of static frames makes it difficult to compare and directly observe the improvements. To address this issue, we plan to revise Fig.3 by using GIFs for each individual video instead of sequences of static frames. This approach will provide a much clearer and more effective comparison. The revised version can be found in Fig.1 of the attached PDF. For the best viewing experience, please use Adobe Acrobat Reader or another PDF reader that supports animation, as browser support for this feature is limited.

---

### Comment · Area_Chair_ZGp6 · 2024-08-08

Dear Reviewers,

Please read authors' responses carefully and provide your answers.

Thanks,
AC

---

### Decision · Program_Chairs · 2024-09-25

**Decision:**

Accept (poster)

**Comment:**

This paper proposes a method to enhance motion dynamics in T2V generation by latent diffusion modeling. In specific, it decomposes both of the text encoder and the text  conditioning module into their respective content and motion components and applies the motion-specific training losses to focus on motion encoding from an input text and motion generation in video. Experimental results show that the proposed model, DEMO, on a number of T2V benchmarks improves performances over baselines in terms of faithful and realist motion generation while maintaining high image quality.

Overall, the motivation is clear and the proposed method seems to be technically sound. In addition, the authors thoroughly address most of concerns and issues raised by the reviewers with more experiments.

However, the description of the proposed method needs to be improved. Especially, it is unclear what modules are optimized by each loss. Mathematical notations and typos also need to be improved (what notation indicates the text conditioning module?) and fixed (Eq. (1), (8)). There are also a number of questions in the proposed method. The main motivation of the proposed decomposition is a lack of motion encoding by CLIP, but L_reg tries to bring the motion encoded text vector and CLIP image vector closer. Also, why does L_video-motion in Eq. (9) not use the optical flow for the motion guidance like L_text-motion in Eq. (5), why do these two motion guidance approaches use different loss formulations?

In empirical validation, it is hard to recognize improved motions by the proposed method from the current examples, as the reviewers said. Moreover, ablation studies are lacking. For example, there is no analysis on the scaling factors of losses, and on performances without L_text-motion or L_reg.

I think this paper is a borderline paper including above weaknesses. However, based on considerable technical contribution and overall positive ratings by the reviewers, I would recommend the paper to be accepted while I would recommend the authors to faithfully revise the paper reflecting comments from the reviewers and me.